# Substantial Obstetric Anal Sphincter Injury during Vacuum Assisted Delivery: An Obstetrical Issue or Device Related?

**DOI:** 10.3390/jcm11236990

**Published:** 2022-11-26

**Authors:** Yoav Baruch, Ronen Gold, Hagit Eisenberg, Hadar Amir, Yariv Yogev, Asnat Groutz

**Affiliations:** 1Urogynecology and Pelvic Floor Unit, Department of Obstetrics and Gynecology, Tel Aviv Medical Center, Faculty of Medicine, Tel Aviv University, Tel Aviv 6997801, Israel; 2Faculty of Medicine, Tel Aviv University, Tel Aviv 6997801, Israel; 3Lis Maternity Hospital, Tel Aviv Medical Center, Sackler Faculty of Medicine, Tel Aviv University, Tel Aviv 6997801, Israel

**Keywords:** OASI, Vacuum, operative vaginal delivery, perineal tear, risk factors

## Abstract

Background: Obstetric anal sphincter injuries (OASIS) might be associated with long-term urinary and anorectal morbidities. The aim of the study was to investigate the risk factors and clinical implications of OASIS associated with vacuum-assisted deliveries versus normal vaginal deliveries. Methods: A series of 413 consecutive OASIS cases were retrospectively analyzed. A comparison was made between OASIS cases diagnosed following vacuum-assisted deliveries versus OASIS cases diagnosed following normal vaginal deliveries. Multivariable analysis was used to study the association between vacuum-assisted deliveries and superficial (3A and 3B) versus deep (3C and 4) perineal tears. Results: The study population comprised 88,123 singleton vaginal deliveries. Diagnosis of OASIS was made in 413 women (0.47% of the total cohort), 379 (91.8%) of whom had third-degree tears and 34 (8.2%) of whom had fourth-degree tears. Among the 7410 vacuum-assisted deliveries, 102 (1.37%) had OASIS, whereas, among the 80,713 normal vaginal deliveries, only 311 (0.39%) had OASIS. In a multivariate analysis, only vacuum-assisted delivery was found to be associated with a significant risk of deeper (3C or 4) perineal tears (OR = 1.72; 95% CI 1.02–2.91; *p* = 0.043). Conclusions: Vacuum-assisted instrumental intervention is a significant risk factor for OASIS and especially for deeper tears, independent of other maternal and obstetric risk factors.

## 1. Introduction

Obstetric anal sphincter injuries (OASIS) are a direct result of obstetric trauma during vaginal delivery. In addition to the immediate pelvic floor injury, OASIS is associated with significant long-term medical and emotional morbidities. The incidence of OASIS in modern obstetric practice varies in different populations, ranging from an incidence of less than 1% to an incidence of 6% [1,2,3,4]. The risk factors for OASIS are similar worldwide and include primiparity, birthweight >4000 g, instrumental assisted deliveries, persistent occiput posterior presentation, a prolonged second stage of labor, median episiotomy, and previous OASIS [4,5,6,7,8,9,10]. In addition, there is controversy about other possible risk factors such as vaginal birth after a cesarean section, maternal age, ethnicity, and more. Both internal and external anal sphincters maintain the anal continence mechanism. The internal sphincter is a continuation of the circular smooth muscle of the gastrointestinal tract and is innervated by the autonomic nervous system. The external sphincter is a striated muscle innervated by the pudendal nerve. The severity of OASIS is determined by the involvement of the anal sphincters and the inner rectal mucosa and is classified into third- and fourth-degree tears. Most studies address risk factors and clinical implications with respect to all types of OASIS, without distinguishing between relatively superficial tears (3A or 3B) and deep tears (3C or 4). Additionally, the relative contribution of the various risk factors, each individually, or in combination, is unclear.

The aim of the present study was to investigate and compare risk factors and clinical implications of OASIS associated with vacuum-assisted deliveries versus normal vaginal deliveries in a large tertiary medical center with more than 12,000 deliveries per year.

## 2. Materials and Methods

This is a retrospective large cohort study of 413 women diagnosed with OASIS following singleton vaginal deliveries, over a 10-year period (January 2011 to December 2020). The study was conducted in a tertiary university-affiliated hospital with over 12,000 deliveries per year. Demographic, clinical, and obstetrical data were retrieved from a computerized database. Exclusion criteria were age < 18 years, stillbirths, and breech presentation. The study protocol was reviewed and approved by the institutional ethical review board. In our institution, physiological vaginal births are managed by midwives, while vacuum-assisted deliveries are performed by obstetricians. Forceps-assisted deliveries are not applied at our institution. Vacuum-assisted deliveries are performed mainly for one of two indications: non-reassuring fetal heart rate (NRFHR) or a prolonged second stage of labor (PSS). A prolonged second stage is defined as two hours or more for nulliparous and one hour or more for parous women, with an additional hour for women under epidural anesthesia. The decision to use intrapartum ultrasound before the instrumental intervention is made by the obstetrician according to clinical considerations. The vacuum device is placed only when the fetal head is stationed below the level of the ischial spines (at +1 or more). During the extraction of the newborn, protection of the perineum is routinely performed. Medio-lateral episiotomy is performed selectively according to the clinical judgment of the obstetrician. Diagnosis and severity of obstetric tears are performed at our institution only by obstetricians. 

The American College of Obstetrics and Gynecology criteria are used to classify third and fourth-degree perineal tears [11]: Third-degree tears are defined as the involvement of the external and/or internal anal sphincter and are classified into 3A (less than 50% of external anal sphincter thickness torn); 3B (more than 50% of external anal sphincter thickness torn); and 3C (both external and internal anal sphincters torn). A fourth-degree tear involves the rectal mucosa. We further divided third- and fourth-degree perineal tears into two categories: superficial (3A and 3B) or deep (3C and 4) OASIS.

Demographic, maternal, obstetric, and neonatal parameters were compared between OASIS cases diagnosed following vacuum-assisted deliveries versus OASIS cases diagnosed following normal vaginal deliveries. Further comparison was made between the two categories of superficial (3A and 3B) versus deep (3C and 4) OASIS. 

Statistical Analysis: Categorical variables were compared between the two main study groups by using the Chi-square or Fisher’s exact tests and are presented as frequency and percentage. Continuous variables and ordinal variables were compared by using the *t*-test or Mann–Whitney tests. Continuous variables with normal distribution are presented as the mean and standard deviation (SD), while skewed variables are presented as a median and interquartile range. Multivariable logistic regression was used to study the association between vacuum-assisted deliveries and deep (3C and 4) degree perineal tears while controlling for possible confounders. All statistical tests were two-sided and a *p*-value of less than 0.05 was considered statistically significant. SPSS software (IBM SPSS statistics, version 27, IBM Corporation, Armonk, New York, NY, USA 2020) was used for all statistical analyses.

## 3. Results

The study population comprised 88,123 singleton vaginal deliveries, of which 7410 (8.4%) were vacuum-assisted deliveries. Diagnosis of OASIS was made in 413 women (0.47% of the total cohort), 379 (91.8%) of whom had third-degree tears, and 34 (8.2%) had fourth-degree tears. A comparison of OASIS cases according to the mode of delivery is presented in Table 1. The overall distribution of third- and fourth-degree tears was similar in both groups of vacuum-assisted and normal vaginal deliveries. However, the incidence of deep (3C or 4) perineal tears was almost double following vacuum-assisted deliveries compared with normal vaginal deliveries (37.2% versus 22.9%; *p* ≤ 0.01).

Of the 7410 women who underwent vacuum-assisted deliveries, 102 (1.37%) had OASIS, whereas, among the 80,713 women who underwent normal vaginal deliveries, only 311 (0.39%) had OASIS. Indications for vacuum-assisted deliveries included NRFHR (N = 42; 41.1%), PSS (N = 37; 36.3%), or both NRFHR and PSS (N = 23; 22.6%). A comparison of women with OASIS according to the mode of delivery is presented in Table 2. Maternal parameters were similar in both groups, except for a significantly higher rate of primiparity among OASIS cases after vacuum-assisted deliveries compared with OASIS cases after normal vaginal deliveries (94.1% versus 64.6%, *p* < 0.001). Incidence of deep (3C or 4) perineal tears among primiparous women was significantly higher following vacuum-assisted deliveries compared with normal vaginal deliveries (36.4% versus 24.3%; *p* = 0.04). 

A comparison of intrapartum obstetric parameters of women with OASIS according to the mode of delivery is presented in Table 3. As expected, there were significantly more cases of PSS among women who underwent vacuum-assisted deliveries. The incidence of deep (3C or 4) perineal tears among cases of PSS was almost double following vacuum-assisted deliveries compared with normal vaginal deliveries (32.7% versus 18.4%; *p* = 0.03). Interestingly, although mean neonatal birth weights were similar in both study groups, the incidence of macrosomia (birth weight ≤ 4000 g) was significantly higher among women with OASIS following normal vaginal compared with vacuum-assisted deliveries (14.7% versus 2.9%; *p =* 0.001). 

After delivery, of all women who underwent vacuum-assisted deliveries, 326 (4.4%) required blood transfusions, preponderantly more so if they endured OASIS (N = 24; 23.5%).

Multivariate logistic regression showed that among all factors found to be statistically significant in the initial comparison, only vacuum-assisted delivery was found to be significantly associated with deeper (3C or 4) perineal tears (OR = 1.72; 95% CI 1.02–2.91; *p* = 0.043).

## 4. Discussion 

Results of the present study show that vacuum-assisted delivery is a robust risk factor for OASIS compared with other well-known risk factors and is significantly associated with deeper (3C or 4) tears and the need for blood transfusions. To the best of our knowledge, this is the first study that compares vacuum-assisted versus normal vaginal deliveries complicated by OASIS. Previous studies have shown that the incidence of OASIS in vacuum-assisted deliveries is higher than in normal vaginal deliveries. In a study conducted in Finland during 2004–2007, the data of 16,802 vacuum births were examined [12]. The incidence of OASIS among nulliparous women was significantly higher in vacuum-assisted compared with normal vaginal deliveries (3.4% versus 1.3%, respectively). Similarly, the incidence of OASIS among multiparous women was significantly higher in vacuum-assisted deliveries compared with normal vaginal deliveries (1.4% versus 0.2%, respectively). In another Scandinavian study, the data of 596 vacuum-assisted deliveries from six different centers in Sweden were examined [13]; OASIS occurred in 71 (11.9%) of the 596 women and there was no correlation between obstetric considerations and management of the instrumental intervention and the risk of OASIS. A recently published Israeli study examined the incidence and risk factors of OASIS in vacuum-assisted deliveries [14]. Among the 9116 vacuum-assisted deliveries during the years 1988–2015, OASIS was diagnosed in only 94 women (1.03%). The main risk factor for OASIS among these women was nulliparity (OR 3.34; 95% CI 1/93–5.78). However, in another recently published Israeli report from a different medical center, vacuum-assisted deliveries were not found to be associated with an increased risk of OASIS regardless of maternal parity status [12,13]. 

Although most data indicate a higher incidence of OASIS in vacuum-assisted deliveries, the mechanism by which this injury is triggered is unclear. Both maternal and obstetric potential confounders limit the ability to understand whether the use of the device itself causes the injury, or whether the indication for the instrumental intervention is the one that leads to the injury. Specifically, it is possible that in cases of PSS the tissues are damaged by the prolonged over-distention, while instrumental intervention due to NRFHR does not allow the pelvic floor tissues to undergo the required physiological adaptation to the fetal head. In addition, it is possible that in nulliparous women compared to parous women, the pelvic floor tissues are less pliable and therefore more exposed to damage during the instrumental intervention. Unlike previous studies that compared vaginal deliveries with versus without OASIS, the current study compared only OASIS cases with versus without instrumental intervention in order to investigate whether the instrumental intervention per se involves an increased risk of OASIS. Additionally, we further divided third- and fourth-degree perineal tears into two categories: superficial (3A and 3B) or deep (3C and 4) OASIS. Of the 7410 women who underwent vacuum-assisted deliveries, 102 (1.37%) had OASIS, whereas, among the 80,713 women who underwent normal vaginal deliveries, only 311 (0.39%) had OASIS. The incidence of deep (3C or 4) perineal tears was almost double following vacuum-assisted deliveries compared with normal vaginal deliveries (37.2% versus 22.9%; *p* < 0.01). All maternal and obstetric parameters were similar in both groups except for a higher incidence of primiparity and PSS among OASIS cases in the vacuum group. There were deeper (3C or 4) perineal tears among primiparous women and cases of PSS delivered with vacuum assistance. However, in a multivariate analysis, only the vacuum intervention per se was found to be associated with a significant risk of deeper (3C or 4) perineal tears (OR = 1.72; 95% CI 1.02–2.91; *p* = 0.043). These data indicate that the instrumental intervention in itself is a strong risk factor for extensive perineal damage compared to other known risk factors and is associated with a significant increase in the rate of deeper OASIS. 

The strengths of this study include the analysis of a large group of OASIS cases in an advanced tertiary obstetric center and a unique comparison between OASIS cases with versus without vacuum assistance allowing the differentiation between the instrumental procedure itself and other risk factors. The main weakness of this study is the overall relatively low rate of OASIS. This low rate has also been consistently reported in other Israeli centers and can be attributed to ethnic differences, advanced delivery methods, cesarean section rates, and possibly, although less likely, the underdiagnosis of OASIS [2,3,14,15,16].

## 5. Conclusions

The results of this study indicate that vacuum-assisted instrumental intervention is a significant risk factor for OASIS and especially for deeper tears, independent of other maternal and obstetric risk factors.

## Figures and Tables

**Table 1 jcm-11-06990-t001:** Comparison of OASIS according to the mode of delivery.

	Normal Vaginal Deliveries (N = 311)	Vacuum-Assisted Deliveries (N = 102)	*p*
Third-degree tears	289 (92.9%)	90 (88.2%)	NS
3A	118 (37.8%)	27 (26.5%)	0.049
3B	96 (30.7%)	32 (31.4%)	0.054
3C	42 (13.5%)	23 (22.5%)	0.049
3 unspecified	33 (10.6%)	8 (7.8%)	NS
Fourth-degree tears	22 (7.1%)	12 (11.8%)	NS
Superficial (3A or 3B)	215 (77.1%)	59 (62.8%)	<0.01
Deep (3C or 4)	64 (22.9%)	35 (37.2%)	<0.01

NS = statistically non-significant, *p* > 0.05.

**Table 2 jcm-11-06990-t002:** Comparison of women with OASIS according to the mode of delivery.

Mean ± SD; or N (%)	Normal Vaginal Deliveries (N = 311)	Vacuum-Assisted Deliveries (N = 102)
Age (years)	30.67 (±4.7)	30.55 (±4.3)
Primiparity	201 (64.6%) *	96.0 (94.10%) *
Comorbidities	18 (5.8%)	6 (5.9%)
Alcohol use	5 (1.6%)	0 (0%)
Smoking	8 (2.6%)	1 (1.0%)
Drug use	2 (0.7%)	0 (0%)
Diabetes	18 (5.8%)	5 (4.9%)
Weight in the first trimester (kg)	59 (±11.6)	59.8 (±12.3)
BMI (kg/m^2^)	22.1 (±3.7)	22.7 (±4.3)
Weight at delivery (kg)	72.4 (±11.9)	73.3 (±12.3)
Pregnancy weight gain (kg)	13.6 (±5.3)	13.9 (±5.0)

* Statistically significant; *p* < 0.001.

**Table 3 jcm-11-06990-t003:** Comparison of intrapartum obstetric parameters of women with OASIS according to the mode of delivery.

Mean ± SD; or N (%)	Normal Vaginal Deliveries (N = 311)	Vacuum-Assisted Deliveries (N = 102)	*p*
Gestational week at delivery	39.8 (39.2–40.7)	40.14 (39.2–41.2)	0.037
Epidural analgesia	200 (64.3%)	91 (89.2%)	NS
Intrapartum fever	8 (2.6%)	4 (3.9%)	NS
Mediolateral episiotomy	71 (22.7%)	72 (70.6%)	<0.001
Occiput posterior position	7 (2.2%)	18 (17.6%)	<0.001
Use of Pitocin	162 (52.1%)	92 (90.2%)	<0.001
Duration second stage (min)	68 (20–139.5)	178 (107.3–216.7)	<0.001
Prolonged second stage	55 (17.6%)	60 (58.8%)	<0.001
Duration third stage (min)	12 (10–17)	13 (10–19)	0.039
Shoulder dystocia	5 (1.6%)	3 (2.9%)	NS
Apgar 5 min less than 7	3 (1.0%)	1 (1.0%)	NS
Birth weight (grams)	3512.7 (±434)	3371.7 (±382)	0.002

NS= statistically non-significant, *p* > 0.05.

## Data Availability

Data supporting reported results can be found in a specific archived database generated during the study.

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
