# Peer review of "Substantial Obstetric Anal Sphincter Injury during Vacuum Assisted Delivery: An Obstetrical Issue or Device Related?"

_jcm, 2022, doi:10.3390/jcm11236990_

Round 1
Reviewer 1 Report
Dear Authors,
I have read with great interest your paper on OASIS and vacuum-assisted delivery.
I have a few remarks below:
1. Do you use intrapartum ultrasound before the decision of operative vaginal delivery?
2. Could you elaborate more on the midwives' competencies in your setting? It might happen that only the midwife assesses the perineal trauma in spontaneous vaginal deliveries, hence underreporting of OASIS in sVD?
3. Could you elaborate on unnecessary episiotomy as a risk factor for OASIS? It seems not only median episiotomy is a risk factor.
https://obgyn.onlinelibrary.wiley.com/doi/epdf/10.1002/ijgo.14101
4. Why the rates of Pitocin use are so high in both groups? Especially in the OVD group- do you routinely use Pitocin to strengthen the contractions during the procedure? When NRFHR is the indication for OVD, Pitocin withdrawal might improve fetal well-being and thus make OVD unnecessary.
5. Is your conclusion after the results of this study that OVD itself should be an indication for episiotomy?
I look forward to your comments,
Best regards
Author Response
Reviewer 1:
We thank the reviewer for the constructive comments. Attached are our responses to the comments and as well as the revised manuscript.
- Do you use intrapartum ultrasound before the decision of operative vaginal delivery?
The decision to use intrapartum ultrasound before the instrumental intervention is made by the obstetrician according to clinical considerations. This is now specified in the Methods section.
- Could you elaborate more on the midwives' competencies in your setting? It might happen that only the midwife assesses the perineal trauma in spontaneous vaginal deliveries, hence underreporting of OASIS in sVD?
Diagnosis and severity of obstetric tears is performed at our institution only by obstetricians, even when it comes to normal vaginal births. Midwives undergo training to identify obstetric tears, but the decision regarding the severity of the tears is made by doctors only. This is now specified in the Methods section.
- Could you elaborate on unnecessary episiotomy as a risk factor for OASIS? It seems not only median episiotomy is a risk factor.
As was specified in the Methods section, we perform medio-lateral episiotomies only, and each episiotomy is performed selectively according to the clinical judgment of the obstetrician. We therefore believe that the number of unnecessary episiotomies is negligible. Further, in the statistical analysis, no significant difference was found between the study groups with regard to episiotomy.
- Why the rates of Pitocin use are so high in both groups? Especially in the OVD group- do you routinely use Pitocin to strengthen the contractions during the procedure? When NRFHR is the indication for OVD, Pitocin withdrawal might improve fetal well-being and thus make OVD unnecessary.
We don’t routinely use Pitocin during vacuum assisted deliveries. Pitocin is administrated only upon obstetric indications. Further, the overall use of Pitocin among normal vaginal deliveries in our institution is less than 15%. It is possible that the high rate of Pitocin among cases complicated with OASIS is associated with the indication for the instrumental intervention, i.e. prolonged second stage.
- Is your conclusion after the results of this study that OVD itself should be an indication for episiotomy?
An excellent question, however, according to the data collected in this study and the statistical analysis, it is not possible to conclude on a connection between episiotomy and the presence or absence of OASIS in vacuum assisted deliveries.
Reviewer 2 Report
Thank you for requesting to provide a review of this article, which has a subject of high interest, as obstetric anal sphincter injuries (OASIS) tend to be a real problem regarding the life quality of the patient and leads to many risks and long term complications.
The main purpose of the analysis was to investigate risk factors and clinical implications of OASIS associated with vacuum assisted deliveries VS normal vaginal deliveries. The study was a retrospective cohort study, over a period of time between January 2011 and December 2020, in which 413 consecutive OASIS cases were retrospectively analyzed, and a comparison was made between OASIS after vaccum assisted deliveries and OASIS after normal vaginal deliveries.
Regarding the structure and accuracy of the phrases, the manuscript has well structured information, with supported evidence and well structured phrases.
The manuscript is original and well defined. The results provide an advance in current knowledge. The results are being interpreted appropriately and are significant, as well as the conclusions.
The article is written in an appropriate way.
The study is correctly designed and the analysis is being performed at high standards, so the data are robust enough to draw the conclusion.
Surely the paper will attract a wide readership.
The English language is appropriate and well understandable.
I only have a few things to add in the lines below:
Line 22: „,” before „whereas”
Line 22: „,” before „only”
Line 46: „,” before „without”
Line 48: „,” after „in combination”
Line 171: „,” before „only”
Line 190: The results of the study..., not „Results of the study...”
Author Response
Reviewer 2:
Thank you for your comments. All requested changes were done accordingly.
Line 22: „,” before „whereas”
Line 22: „,” before „only”
Line 46: „,” before „without”
Line 48: „,” after „in combination”
Line 171: „,” before „only”
Line 190: The results of the study..., not „Results of the study...”